Exploring the impact of positive and negative emotions on cooperative behaviour in a Prisoner’s Dilemma Game

Kjell Oscar N.E. 1 2 oscar.kjell@psy.lu.se
Thompson Sam 3
1 Department of Psychology, Lund University , Lund , Sweden
2 Department of Psychology, University of East London , London , United Kingdom
3 Institute of Psychology, Health and Society, University of Liverpool , Liverpool , United Kingdom
Iacoboni Marco
Electronic publication date: 2013 Dec 19
Publication date: 2013
Volume: 1
Electronic Location ID: e231
Received 2013 Nov 6; Accepted 2013 Dec 4
Copyright: © 2013 Kjell and Thompson
Copyright year: 2013
Copyright holder: Kjell and Thompson
License: This is an open access article distributed under the terms of the Creative Commons Attribution License, which permits unrestricted use, distribution, and reproduction in any medium, provided the original author and source are credited.
License URL: https://creativecommons.org/licenses/by/3.0/

Keywords: Well-being, Positive and negative emotions, Cooperation, Social dilemma games

Funding: This work was carried out without any funding sources.

==============================
Objective. To explore the influences of discrete positive and negative emotions on cooperation in the context of a social dilemma game.

Design. Two controlled studies were undertaken. In Study 1, 69 participants were randomly assigned to an essay emotion manipulation task designed to induce either guilt, joy or no strong emotion. In Study 2, 95 participants were randomly assigned to one of the same three tasks, and the impact of emotional condition on cooperation was explored using a repeated Prisoner’s Dilemma Game.

Results. Study 1 established that the manipulation task was successful in inducing the specified emotions. The analysis from Study 2 revealed no significant main effects for emotions, in contrast to previous research. However, there was a significant effect for participants’ pre-existing tendency to cooperate (social value orientation; SVO).

Conclusion. Methodological explanations for the result are explored, including the possible impact of trial-and-error strategies, different cooperation games and endogenous vs exogenous emotions.

The Influence of Emotions on Cooperation

The tendency to cooperate (or not) has often been understood as a dispositional trait varying between individuals (Messick & McClintock, 1968). For example, social value orientation (SVO), defined as individual preferences for specific distributions of outcomes between oneself and another (McClintock, 1972), significantly predicts cooperation in social dilemmas (Balliet, Parks & Joireman, 2009).

However, it has more recently been suggested that emotions experienced immediately prior to a cooperative activity may have an impact on the extent of resulting cooperation. Hertel et al. (2000) examined the intuitive assumption that positive emotions increase cooperation. Although they did not find direct evidence to support this hypothesis, they identified that positive and negative emotions, in general, affected decision-making processes, which are known indirectly to affect cooperation depending on the situation. Specifically, participants induced with positive emotions tended to follow specified social norms to a greater extent. However, the study did not consider qualitative differences between discrete emotions, nor individual differences in SVO.

Subsequent studies accounting for these factors have illustrated varied impacts of emotions on cooperation. Ketelaar & Au (2003) measured cooperation in a well-known social dilemma, the Prisoner’s Dilemma Game (PDG). In a two person PDG players have two choices, to cooperate or to defect, with the outcome contingent on the combination of behaviours from each player. What defines the game as a PDG are the relative values of these four possible outcomes (Kollock, 1998). These are calibrated such that the highest outcome for an individual player is produced in instances where s/he defects (i.e., does not cooperate) whilst the other cooperates. The next highest outcome is when both cooperate, followed by when both defect. The lowest outcome is the reverse of the highest, such that the individual cooperates whilst the other player defects. Depending on the scenario in which the PDG is framed, the calibration may be set such that mutual cooperation yields the highest combined outcome of all four combinations. The fundamental “dilemma” of the game, however, is always the same: that the rational strategy for both individuals is to defect, but that this leads to a sub-optimal outcome compared with the situation had they both cooperated.

In order to account for SVO, participants in Ketelaar and Au’s (2003) study undertook 40 repeated rounds of a PDG, which established their tendencies to cooperate or defect. Subsequently, one group of participants was given an emotion induction task in which they wrote about a personal event associated with guilt, whilst the others described a normal day as neutrally as possible. Afterwards, participants engaged in another 40 rounds of the PDG in order to assess any influence of the induced emotion. Results suggested that those participants who initially behaved uncooperatively (i.e., exhibiting a “proself” SVO) cooperated to a greater extent subsequent to writing about guilt compared with those writing about a neutral experience. Importantly, this pattern was not seen in those participants who initially behaved relatively cooperatively (i.e., a “prosocial” SVO). However, the apparent impact of guilt was only seen in the first 10 rounds of the game subsequent to the writing task. Meanwhile, influences of the SVO were seen throughout all 40 rounds.

The same pattern has been replicated in subsequent studies investigating additional discrete emotions. de Hooge, Zeelenberg & Breugelmans (2007) employed a similar emotion manipulation task, with the addition of a shame condition. Cooperation was assessed by means of a give-some dilemma game (GSDG). Participants were paired and given 10 coins (each worth €0.50); however, they had the option of giving their partner some of their coins in order to double their value. Thus, the level of cooperation was measured in terms of how many coins each participant gives away, in just one interaction. SVO was assessed using the Triple-Dominance measure (VanLange et al., 1997). The results supported Ketelaar and Au’s (2003) finding regarding the impact of guilt on cooperation; however, there was no significant effect in terms of shame. Employing a similar methodology, Nelissen, Dijker & deVries (2007) compared guilt and fear. Induced guilt increased cooperation in those classified as proself, whereas induced fear reduced cooperation in those classified as prosocial. As well as adding to the evidence for an impact of guilt on cooperation, this study also illustrates the importance of specific qualitative differences between discrete emotions.

Positive emotions and cooperation within the broaden-and-build theory

To date, discrete emotions considered positive have not been investigated using these kinds of methodologies. Positive emotions have recently been the subject of considerable attention within psychology, with Fredrickson (1998) arguing that positive emotions have a functional utility beyond merely feeling pleasant such as facilitating and building social connections and relationships. That is, within Fredrickson’s broaden-and-build theory, negative emotions are seen to narrow an individual’s range of responses in terms of thoughts and actions. This, in turn, focuses resources on self-protective, “fight” and “flight” behaviours. Positive emotions, meanwhile, are seen to broaden an individual’s range of thoughts and actions. Fredrickson (2001) holds that this temporary broadening facilitates and promotes opportunities for an individual to both discover and further develop enduring personal resources, as well as building future social relationships. Moreover, she argues that this can generate upward spirals of positive emotions, cognitions and actions, allowing for personal development and transformation. Instinctively, this is an attractive theory that provides explicit predictions, some of which have been supported by experimental evidence (see, e.g., Fredrickson & Branigan, 2005). This will be further discussed below in relation to the discrete emotion joy.

Interpersonal aspects of guilt and joy

In the research described above, guilt is commonly defined as a moral emotion, experienced subsequent to a moral transgression. Tangney, Stuewig & Mashek (2007) argue that guilt is a “self-conscious emotion that is evoked by self-reflection and self-evaluation” (p. 347), and subsequently has the potential to promote constructive and proactive pursuits, often towards others. Evidence for the role of guilt in interpersonal relationships comes from an experiment involving functional magnetic resonance imaging (fMRI) scans. Takahashi et al. (2004) found greater brain activity in the areas associated with theory of mind (ToM) and social cognition in participants who read sentences associated with guilt compared with those reading more neutral sentences.

The impact of positive emotions on interpersonal relationships is less clear. Often considered one of the most basic positive emotions (Oatley & Johnson-Laird, 1987), joy is perhaps the archetypal positive emotion and plays a central role in Fredrickson’s broaden-and-build theory. Johnson & Fredrickson (2005) found that joy reduced own-race bias, hypothesizing that the experience of joy may have the effect of increasing a sense of common in-group identity. However, joy has also been defined as a “self-directed or noninterpersonal” emotion derived from pleasure-related experiences (Berenbaum, 2002). Accordingly, Takahashi et al. (2008) compared participants reading sentences associated with joy, pride or a more neutral situation in a fMRI study. They found that pride elicited greater activation in areas associated with ToM (as with guilt), compared with joy and more neutral sentences. Joy, on the other hand, illustrated greater activation in areas linked with the “processing of hedonic or appetitive stimuli” (p. 898) rather than areas associated with interpersonal social stimuli. This would seem to challenge the idea of joy as promoting social cooperation. However, considering the study by Hertel et al. (2000) in which general positive affect facilitated decision making consistent with social norms, it could be hypothesized that joy will not influence cooperation per se, but rather affects cooperation indirectly by increasing self-absorbed/self-directed embracing of social norms (or values).

Hypotheses

The aim of the present research was to build upon previous findings on the influences of discrete emotions on cooperation within the context of a social dilemma. Specifically, it aimed to compare the differential influence of joy and guilt on participants’ inclinations to cooperate in a PDG.

Based on the foregoing arguments it was hypothesized that results from previous studies on guilt and cooperation would be replicated. With respect to joy, our study was exploratory since, as discussed above, the likely direction of impact of joy on cooperation was not clear from existing theory and research.

We report two studies. Study 1 was a validation conducted in order to establish whether an adapted version of the essay manipulation task (e.g., Ketelaar & Au, 2003) was able to reliably induce the discrete emotions of guilt and joy. Study 2 reports results from a study comparing the impact of evoked joy and guilt on subsequent performance in a PDG.

Study 1

Methodology

Participants

Sixty-nine people took part in the study. All were students at the University of East London and ranged between 18 and 51 years of age (mean 22.62). 22 were male and 47 female. In return for participation they were offered an opportunity to enter a prize draw to win a £25 voucher. Both studies reported within this article were approved by the University of East London’s Review Board.

Intervention

The emotion manipulation essay questions were adapted from several previous research studies. Participants were required to write for approximately 10 min about a specific event in their life in as much detail as possible. The control and guilt conditions were adapted from studies carried out by Ketelaar & Au (2003), de Hooge, Zeelenberg & Breugelmans (2007), and Nelissen, Dijker & deVries (2007). In the control condition participants were asked to write about their “regular activities and schedule”, describing events neutrally. In the guilt condition participants were asked to write about an experience that had caused them to feel “regretful and remorseful”, focusing on what made them feel particularly guilty, e.g., cheating or forgetting a friend’s birthday.

Since no previous study had used this methodology to induce joy, sentences were adapted from those used in the fMRI experiment by Takahashi et al. (2008). Hence, participants were asked to write about a personal experience in which they felt “joyful and delighted”, focusing on what made them feel significant pleasure, e.g., receiving a valuable gift or having a delicious dinner.

Measures

Participants’ emotional state post writing was assessed using two instruments: PANAS-X, and the State Shame and Guilt Scale (SSGS).

The PANAS-X (Watson & Clark, 1994) is a 60-item schedule assessing feelings and emotions experienced at the present moment. Responses are recorded along a rating scale ranging from 1 (very slightly or not at all) to 5 (extremely). Subscales representing specific emotions can be calculated from scores on the PANAS-X. In the present study, the dimension “Joviality” (including 8 items such as “happy” and “joyful”) and “Guilt” (6 items including “guilty” and “blameworthy”) were used as dependent measures.

The State Shame and Guilt Scale (SSGS; Marschall, Sanftner & Tangney, 1994) is a 15-item questionnaire developed to measure and distinguish between shame, guilt and pride in the present, based on phenomenological descriptions. Responses are recorded along a Likert type scale ranging from 1 (“Not feeling this way at all”) to 5 (“Feeling this way very strongly”).

In addition, a manipulation check was implemented to enable analysis of the nature of events recalled. This was operationalized using an adapted version of an Essay Evaluation Measure (EEM; Dickerson et al., 2004). This manipulation control questionnaire assesses participants’ own perceptions of the written essay, asking them to judge the extent to which it is personal, emotional and meaningful on a 7-point scale (from “not at all” to “a great deal”). An additional, condition-specific question was added requiring participants to identify how accurate they consider their descriptions (in the control condition), how much they blamed themselves (in the guilt condition), and how much they enjoyed the experience (in the joy condition) at the time that it occurred.

Procedure

Students were approached after lectures, and asked to participate in a short psychology study that involved anonymously writing about a previously experienced event and answering a few questionnaires. Upon showing interest, participants were escorted to a quiet area where they read and completed a consent form.

Participants were randomly assigned to one of the three conditions. All necessary materials were provided at the start of the experiment and they were asked to complete the tasks in the following order: (1) the manipulation task; (2) PANAS-X; (3) SSGS; (4) EEM. Having completed the task and the questionnaires, each participant was debriefed.

Results

Prior to the main analysis, data from the EEM were reviewed. A between-group ANOVA found a statistically significant effect of emotion manipulation condition (F[2, 70] = 14.96, p < 0.005). Post-hoc tests showed that participants in the guilt condition rated their essays significantly higher on the EEM (i.e., more personal, meaningful and emotional) compared with participants in both the joy and control conditions. However, participants in the joy condition also rated their essays significantly higher on the EEM than those in the control condition.

Descriptive statistics for data on self-reported emotional state are given in Table 1. A MANOVA was conducted with emotion manipulation condition (guilt, joy or neutral) as a between-group variable and the PANAS-X Guilt, PANAS-X Joy, SSGS Guilt and SSGS Shame as dependent variables. In terms of missing values, the mean value of the existing data in the relevant dimensions for the specific participant, was subsequently added to the missing fields.

Table 1 Descriptive statistics indicating the mean, standard deviation and frequency of missing values for each condition (N = number of participants), and for the dependent variables (DV).

Condition	DV	
	PANAS-X Guilt	PANAS-X Joy	SSGS Guilt	SSGS Shame	
Guilt
(N = 23)	16.47 (±6.49)
2	21.22 (±9.41)
3	12.78 (±6.35)
0	9.83 (±3.90)
0	
Control
(N = 23)	10.99 (±5.68)
1	24.00 (±7.59)
1	10.52 (±5.20)
0	8.22 (±3.20)
1	
Joy
(N = 23)	8.90 (±4.83)
2	29.17 (±7.73)
1	7.74 (±3.82)
0	6.35 (±2.31)
0	

There was a significant main effect of emotion condition at the p < 0.05 level across the combined dependent variables (F[8, 128] = 3.8) and for each of variables individually (PANAS-X Guilt F[2, 66] = 10.8; PANAS-X Joy F[2, 66] = 5.46; SSGS Guilt F[2, 66] = 5.38; SSGS Shame F[2, 66] = 6.78).

For the PANAS-X Guilt variable, post-hoc tests (Tukey HSD) indicated that participants in the guilt condition reported significantly higher guilt compared with both control and joy conditions (p < 0.05). Participants in the joy condition did not differ significantly from those in the control condition.

For the PANAS-X Joy variable, participants in the joy condition reported significantly more joy compared with participants in the guilt condition. The difference between participants in the joy and control conditions was not significant at the p < 0.05 level. However, it was in the expected direction and was approaching significance (p = 0.094); subsequent analysis revealed that it would have attained significance with a less conservative test (i.e., LSD instead of Tukey). Participants in the guilt condition did not differ significantly from those in the control condition.

On both the SSGS Guilt and SSGS Shame, participants in the guilt condition reported significantly higher levels of both guilt and shame at the p < 0.05 level, compared with participants in the joy condition. However, they did not differ significantly with participants in the control condition.

Discussion

Those participants writing about guilt or joy evaluated their essays as significantly higher on the EEM compared with those writing about a more neutral event. In terms of content, participants in the guilt condition tended to write about experiences in which they, for example, had cheated on a test or a partner, or let down a close friend. Participants in the joy condition, by contrast, wrote about experiences such as receiving an expensive present or winning something. In the neutral condition most participants wrote about normal, everyday activities. Generally, then, we can be confident in participants’ compliance with the manipulation task.

In terms of participants’ subsequent emotional state, scores on the two dimensions of guilt and joy within PANAS-X clearly indicate that the emotion manipulation task influenced participants. As hypothesized, the guilt condition elevated scores on the PANAS-X Guilt scale relative to both the control and the joy conditions. Similarly, participants in the joy condition reported significantly more salient experiences on the joy dimension compared with those participants in the guilt condition. Although this group did not differ significantly from the control condition at the p < 0.05 level, nevertheless there was evidence that the intervention was working as expected in terms of significantly discriminating between the main conditions guilt and joy; albeit perhaps not as strongly as the guilt intervention.

In addition to the PANAS-X, the SSGS measure indicated that participants in the guilt condition reported both significantly more guilt and shame as compared with the joy condition but not the control condition. This suggests that both emotions were present within the guilt condition. It is worth noticing that the lack of difference between the guilt and control condition might be because the joy condition involved decreasing reports or the control condition involved increasing reports of guilt and shame; and that this might illustrate that a control condition should not be considered as totally neutral. (As an interesting aside, this result suggests that contra the studies of de Hooge and colleagues (2007; 2008) it may in fact be difficult to induce guilt in isolation from shame.)

Overall, the results of Study 1 supported use of modified emotion manipulation essay as an intervention to induce positive feelings of joy as well as guilt. However, given the apparent difference in “strength” between the two induction tasks, it was decided to modify the writing tasks slightly for use in Study 2 by adding a sentence to the instructions for both the guilt and joy conditions:

“Try to at least write 10 lines. If you find it hard to remember the details of the event, try to imagine yourself in the situation again”.

The intention was that this would further increase participants’ vividness of recollection and absorption in the task, thereby increasing the strength of emotional induction.

Study 2

Methodology

Participants

As in Study 1, participants were students at the University of East London and recruited in the same manner. However, as a further incentive to participate and to encourage serious engagement in the study, participants were given the opportunity to enter a prize draw to win a £50 voucher. One participant withdrew from the experiment due to other commitments and one participant was excluded for failing to correctly complete one of the questionnaires. The final sample consisted of 95 participants (41 males and 54 females). Participants were between the ages of 18 and 50 years with a mean age of 24.16 years (SD ±5.19).

Interventions

The same manipulation task was used as in Study 1, but with the small modification to the instructions described above.

Measures

SVO was assessed with the Triple-Dominance measure (TDM; VanLange et al., 1997). The TDM involves nine items, each consisting of different distributions of “valuable points” between oneself and a hypothetical unknown other. Each item covers three different distributions, which are normally categorized into two groups: prosocial (equal division) and proself (individualistic and competitive divisions). Participants can be classified as either prosocial or proself according to whether six or more consistent answers fall within one of the two specified categories.

The main dependent variable was frequency of cooperative interactions in 10 rounds of a Prisoners Dilemma Game (PDG). This was facilitated using a specifically developed computer program in which participants engaged in an adapted version of the PDG. The program was developed to mimic a real world interaction between two participants, such that participants believed they were interacting with another person at a different location (see Procedure, below, for more details). Participants could opt to either cooperate or defect whilst the computer program was set to respond according to the “Tit-for-Tat” strategy (in which the initial response involves cooperation and in subsequent rounds matches the participant’s previous mode of response). The scenario for the PDG was described in terms of varying quantities of lottery tickets. The pay-off matrix (see Table 2) corresponded to standard PDGs (Kollock, 1998). After the PDG had been completed, emotional state was measured using the joy and guilt subscales of the PANAS-X. The EEM was again used to enable a manipulation check.

Table 2 Pay-off matrix applied within the current Prisoner’s Dilemma Game.

		Person 2	
		Cooperate	Defect	
Person 1	Cooperate	2,2	0,3	
Defect	3,0	1,1	

Procedure

Participants were told that the experiment involved writing about a previous life event, followed by a computer based interaction with another person and the completion of several questionnaires.

After participants had given consent, the experimenter excused himself, informing them that he had to contact a colleague in another campus in order for them to start the experiment simultaneously. This deceit was intended to further facilitate a perceived sense of interaction during the computer-based experiment. The experimenter made certain to “contact his colleague” using a mobile telephone, in full view of the participants.

Participants were then seated in front of a computer and told that they could potentially increase their chances of winning a £50 voucher depending on the results of their interaction. The first part of the experiment was computer based and involved describing and demonstrating the PDG task in order to familiarise participants. Rules of the interaction were thoroughly explained, and participants were required to answer hypothetical questions about each of the four possible cooperation/defect scenarios. Crucially, they were asked to judge the quantitative distributions between each hypothetical individual, for each scenario presented on the screen. They could not proceed until answering all of the questions correctly. The instructions and rules were written so as to be value neutral, hence rather than using words such as cooperation and defect, participants chose either “Option A” or “Option B”. This was followed by a 10 min countdown in which participants were instructed to complete the demographic questions and the emotion manipulation task. The interaction task began subsequently. Once the participant selected “Option A” or “Option B”, there was a delay, randomly ranging from 7 to 15 s, followed by the response of “the other person”. This was always followed by on-screen feedback stating how many tickets each “player” had received. After 10 rounds of the game, the final screen requested the participant to complete the remaining questionnaires in the following order: PANAS-X, EEM and the TDM questionnaire (note that this was administered last so as to avoid influencing behaviour on the PDG through any kind of priming mechanism).

After completing the questionnaires, participants were debriefed. As in Study 1, participants were briefly informed of the requirements of the study prior to participation and asked to give consent. However, this experiment involved deceiving participants both in terms of the description of the interaction game and the experimenter’s staged “phone call”. The true nature of the study was disclosed sensitively to the participants during the debriefing session; a written debriefing was also provided.

Results

The frequency of cooperative responses was calculated as the linear sum of responses from all 10 rounds of the PDG. Analysis was then undertaken in two stages. Firstly, as in Study 1, data were checked in terms of compliance within the emotion manipulation task. Secondly, there was an investigation of cooperation wherein an ANCOVA was applied to investigate cooperation in all 10 rounds.

Initial control analyses – EEM

Participants’ self-reported retrospective accounts of the emotion manipulation task were analysed as in Study 1. The condition specific items in the EEM indicated that participants in the guilt condition reported their essays as involving self-blame at the time events occurred (mean 5.2, SD ±1.6). Those in the control condition indicated that their account was accurate (mean 5.3, SD ±1.2) and those in the joy condition reported that their essays involved enjoyment (mean 6.0, SD ±1.3). Moreover, combining the scores from the three EEM questions and applying an ANOVA revealed a significant effect for emotion manipulation condition (F[2, 108] = 21.51, p < 0.005). Post-hoc tests showed that participants in the guilt (p < 0.005) and joy (p < 0.005) condition rated their essay as more personal, meaningful and emotional compared with the control condition, but that they were no different from one another in these respects.

Impact of emotional condition on cooperation

To assess the influence of the emotion condition on cooperation over the 10 rounds of the PDG a one-way ANCOVA was calculated with total cooperation as the dependent variable, emotional condition as the independent variable and SVO as a covariate. The main effect of condition was not significant (F[2, 91] = 0.366, p < 0.69); however, the effect of the covariate was significant (F[1, 91] = 9.65, p < 0.005). This suggests that any effect on cooperation due to the emotion condition was small in comparison with the effect of participants’ SVO.

This result was contrary to our hypothesis, based on previous research, that emotional manipulation would have an effect on cooperation over and above SVO. To explore this result further, we used scores on the TDM to assign each participant as prosocial or proself. Out of a total of 95 participants, 41 were classified as prosocial and 37 as proself (17 could not be categorised as either prosocial or proself and were thus excluded from this analyses). Crossed with the three experimental conditions this yielded six groups of approximately equal size (all with N between 11 and 15).

Figure 1 shows the mean percentages of cooperative responses for each group, plotted against each of the 10 rounds. Visual inspection suggests a rather high fluctuation over the first five rounds, but with a pattern beginning to emerge from round six onwards that seems to be influenced by SVO.

Figure 1 Cooperation in % for each round of the PDG.

Figure shows the mean percentages of cooperative responses for each group, plotted against each of the 10 rounds. Visual inspection suggests a rather high fluctuation over the first five rounds, but with a pattern beginning to emerge from round six onwards that seems to be influenced by SVO.

To explore whether there was a significant “first round” effect, logistic regression was used to estimate the likelihood of either cooperation or defection in the first round, with emotion manipulation condition and SVO as independent variables. A total of 68 cases were analysed and resulted in the full model approaching significance in predicting cooperation (omnibus chi-square = 10.87, df = 5, p = 0.054). The model accounted for between 10.8% and 14.5% of the variance in terms of cooperation, with 46.3% of the cooperative responses predicted, whilst 74.1% of the defective responses were predicted. In total, 62.1% responses were predicted accurately. Table 3 contains coefficients and Wald statistics as well as associated degrees of freedom and probability values for each of the predicting variables and their interaction. This reveals that emotion condition, SVO and their interaction predicted cooperation in the first round.

Table 3 Values of the predicting variables extracted from a logistic regression analysis.

Predicting
variables	Coefficient	Wald	Degree of
freedom	Probability
value	
Condition	−2.23	5.78	1	.016	
SVO	−2.26	6.51	1	.011	
Interaction	-	7.73	2	.021	

Discussion

As in Study 1, the EEM suggested that participants perceived that the content of their essays met the requirements of the specified emotional conditions. Participants writing about guilt or joy rated their essays as significantly more personal, meaningful and emotional as compared with the control condition. In addition, participants in Study 2 spent the same amount of time writing whilst also being subject to the enhanced instructions regarding length and increased imagination. Further, informal review of the content of the essays suggested that participants also tended to write about similar experiences as those in Study 1. In the joy condition, these involved detailed accounts of experiences such as engaging in pleasurable activities, getting new things and winning prizes as well as travelling and being with friends; compared with accounts of neglecting, disappoint and letting down friends, family and partners as well as cheating in tests in the guilt condition. Considering these observations together with the evidence from Study 1, and also with the fact that several of the studies discussed earlier have utilised the same induction task, it felt reasonable to assume that participants adhered to the manipulation requirements. Thus, they could be construed as experiencing the specified emotion at the time when they undertook the PDG.

The main analysis suggested, contrary to our hypothesis, frequency of cooperation was not influenced by the emotion manipulation condition. However, the significance of the SVO as a covariate suggests that participants’ existing tendency towards cooperative behaviour was outweighing any impact of the emotion induction. Our subsequent exploratory analysis of progression through all 10 rounds sheds some additional light on this. Regressing responses to the first interaction on emotional condition suggested that condition tended to predict the likelihood of a cooperative response. However, we would interpret this with caution, since it is evident from Fig. 1 that the first five rounds were characterised by a wide range of mean responses, whilst the latter five interactions appeared – at least from visual inspection of the data – to be more strongly influenced by SVO. That SVO has a strong influence on cooperation is consistent with previous research (Balliet, Parks & Joireman, 2009). However, it raises the question of the extent to which momentary emotional states can achieve the same, at least in the long term. It seems, in the present data at least, that the effect of emotion manipulation on cooperation dispersed quickly, existing only within the first round, if at all.

General Discussion

The current results support the notion that cognitive resources and strategies (operationalised as SVO) play an important role in cooperation; this is consistent with an extensive recent meta-analysis (Balliet, Parks & Joireman, 2009). In terms of the overall aim of the study, however, we are not able to draw clear inferences about the effect of positive emotions on cooperation, since our results found no impact of either joy or guilt on cooperation in a PDG. Evidently there are discrepancies between the results from the current study and previous research. It is therefore worth exploring some of the possible reasons why our results did not follow the same pattern even in relation to guilt.

Clearly, it is possible that some procedural flaw gave rise to the observed result whereby there was no discernible impact on cooperation of emotional condition over and above SVO. However, each aspect of the methodology – the emotion manipulation task, the cooperation game, the measure of SVO and the procedural order – were drawn directly from previous research. The experimental procedure was carefully piloted and tested. Debriefing participants revealed that the element of deception (i.e., playing against a “real” opponent in a different campus) had been successful, and no systematic problems were identified.4

Assuming that our methodology “worked” as intended, and that findings from previous studies are robust, what could explain the present result? Two possibilities, not mutually exclusive, are that some aspect of the procedure rendered the effect of momentary emotion on cooperation small in comparison with that of individuals’ existing predisposition towards cooperation, and that aspects of the procedure in previous studies artificially magnified what are in reality very small effects. What aspects of our procedure might be culpable?

The apparent irregularities throughout rounds 1 to 5 might suggest that participants did not take the game seriously, perhaps due to the low stakes involved. However, previous studies have identified effects in instances where stakes varied considerably, from lottery tickets (Nelissen, Dijker & deVries, 2007), to expected pay-offs of $2 each (Ketelaar & Au, 2003) to €20 each (de Hooge, Zeelenberg & Breugelmans, 2007). Moreover, our interpretation of the pattern of response shown in Fig. 1 is that the initial irregularity reduced markedly after five interactive rounds. If this is correct, it seems unlikely that participants would initially choose an option carelessly but start taking the game more seriously after a few rounds.

Perhaps a more likely explanation is that differences in the type of dilemma game employed, and the number of plays, may encourage participants to employ different strategies. For instance, Traulsen et al. (2009) identified a “common” but rarely researched phenomenon, namely that individuals tend to “imitate more successful strategies but sometimes also explore the available strategies at random” (p. 709). Helbing et al. (2005) add that in this way, actors learn to acquire a strategy through the course of interactions. Hence, they argue that seemingly random, exploratory strategies can appear to result in increased cooperation. Accordingly, Ren, Wang & Qi (2007) point out that “the mechanism of randomness promoting cooperations resembles a resonance like fashion, wherein the randomness-induced prevalence of the ‘good’ strategy, i.e., cooperations, evokes the positive effect of the …randomness on the system” (p. 2). The suggestion, then, is that participants are initially more likely to explore different options, in order to subsequently settle on what they believe is the most successful strategy for both themselves and the situation.

With this in mind, it is notable that within the Ketelaar & Au (2003) study participants had already interacted for 40 rounds before the emotion manipulation tasks. This was done so that SVO could be inferred from participants’ pattern of responses; however, it may have also afforded them the opportunity to acquire strategic experience. Thus, when they started to interact again they may have been less likely to explore their options using a trial-and-error strategy — in turn, this may have served to amplify the small effect of emotion manipulation by reducing the proportion of variation in responses attributable to (trial and) error. In contrast, within the current game participants came straight from the interventions to play the PDG for the first time. The game had been carefully explained to them and their understanding had been checked through a series of questions, but they had not actually played it. Further research might beneficially investigate individual differences in trial-and-error strategies, rather than only attempting to counteract them.

Another explanation derives from Nelissen, Dijker & deVries (2007), who note in passing that their use of a one-shot GSDG (as opposed to a repeated PDG) was designed to eliminate trial-and-error responses (a similar methodology has been utilised by de Hooge and colleagues, 2007; 2008). A further beneficial feature is that it lends itself to interval level data (i.e., participant’s choose to give 0 to 10 coins) and thus the application of a more powerful analysis. In the present study, only categorical data were available for each round of the PDG (i.e., cooperate or defect). The logistic regression conducted on the first round of data effectively treated the PDG as a “one shot”. However, the results were inconclusive, and it is probably the case that a binary outcome is insufficiently sensitive to identify a small effect of emotional manipulation.

A final explanation for the divergence of current findings from those reported in previous studies is the nature of the manipulation task. It has previously been shown that the influence of induced emotions diminishes when individuals become aware that the cause of their emotions is exogenous (that is, unrelated) to the task at hand (Higgins, 1996). In their study using repeated PDG, Ketelaar & Au (2003) found exogenous guilt to have an impact in only the first 10 of 40 rounds; however, in a second study, endogenous guilt (i.e., guilt that participants experienced from not cooperating within the social dilemma game itself) was found to increase cooperation one week later in a GSDG. Moreover, de Hooge, Breugelmans & Zeelenberg (2008) found that endogenous and exogenous shame possess different qualitative characteristics, so that endogenous shame seems to increase cooperation whilst exogenous shame does not. Hertel et al. (2000) suggest that “whereas trial and error strategies or deciding by chance might be an appropriate heuristic behaviour when no useful cues are at hand, this situation should change when valid cues are available” (p. 445). Thus, it could be argued that participants in the current study quickly realized the exogenous nature of their emotions and therefore after the first round, engaged in trial-and-error strategies before their SVO took over. Crucially, however, although previous studies indicate the importance of excluding other strategies, we cannot be certain that the first round was not the result of trial-and-error.

Conclusion

The present study provides further evidence for the importance of SVO as a determinant of behaviour in cooperation. However, it did not support the findings of previous research of a consistent impact of experienced guilt on cooperation over and above SVO, nor did it show any impact of joy on cooperation.

More research is required in order to develop a full understanding of discrete emotions’ impact on cooperation. As well as giving systematic consideration to the impact of different experimental methodologies (e.g., repeated PDG vs. one short GSDG), such research should acknowledge the importance of distinguishing between exogenous and endogenous emotions and perhaps compare their respective influences and interactions with existing SVO.

We want to thank Tony Leadbetter for programming the Prisoner’s Dilemma Game quickly and efficiently as well as Dr Joan Painter for her support and advice.

Additional Information and Declarations

Competing Interests

Author Contributions

Human Ethics

4 It may also be that previous results overstate the real effect of emotion on cooperation due to the impact of publication bias and/or false positives. This possibility that must be taken seriously in light of recent discussions, especially in social psychology (see, e.g., Simmons, Nelson & Simonsohn, 2011).

The authors declare that they have no competing interests.

Oscar N.E. Kjell conceived and designed the experiments, performed the experiments, analyzed the data, contributed reagents/materials/analysis tools, wrote the paper.

Sam Thompson conceived and designed the experiments, analyzed the data, contributed reagents/materials/analysis tools, wrote the paper.

The following information was supplied relating to ethical approvals (i.e., approving body and any reference numbers):

The University of East London’s ethics committee approved the study design as part of the main author’s dissertation for the Master in Applied Positive Psychology. The review process was anonymous to ensure objectivity. The overall aim was to ensure that the students’ study designs adhered to APA’s ethical guidelines of psychologists’ code and conduct. Without the ethical approval one could not complete the Master dissertation.

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
