# Peer review of "Exploring the impact of positive and negative emotions on cooperative behaviour in a Prisoner’s Dilemma Game"

_PeerJ, doi:10.7717/peerj.231_

## Round 0.1 · original submission · Minor Revisions

As you can see, both reviewers have been very positive. Dealing with the minor issues raised by reviewer 2 will definitely improve the quality of an already good manuscript.

·

Basic reporting

The paper deals with the impact of positive and negative emotions (joy and guilt) on cooperative behavior, examined using a neuroeconomic game, called the Prisoner’s Dilemma (PDG). Participants were instructed to provide short stories about personal events in which they had experiences joy, guilt or a neutral emotion. The task was subsequently modified, before participants were asked to play a PDG. In addition, social value orientation was measures. Unexpectedly, neither joy nor guilt impacted the participants’ behavior in the PDG, whereas social value orientation did. Several possible explanations are discussed why emotions did not influence cooperation in the PDG.

Overall, the paper is well written. The Introduction covers much of the relevant material. Methods and Results are adequately described. Statistics are sound.
The Discussion comprises several explanations for the negative results of this study (with regard to current emotions). The study sheds some light on the efficacy (or the lack thereof) of psychological manipulation prior to economic decision-making.

Experimental design

Adequate.

Validity of the findings

good.

Additional comments

As above.

·

Basic reporting

The statement made in Lines 73 and 74 hints at an interesting and important idea but is far too vaguely stated. We recommend elaborating on it or rephrasing.

Experimental design

In the discussion of the first study, there is a point (214-216) in which a more thorough explanation or evidence is necessary to accept the authors' conclusion that "although this group did not differ significantly from the control condition...there was evidence that the intervention was working as expected".

In both discussion sections, the evidence of participants' adherence to the manipulation requirements seems relegated solely to informal reviews. Given the importance of the manipulation to interpreting their findings, the authors should provide more details regarding subjects' writings.

Validity of the findings

Regarding study 1 It is notable that participants's scores on the SSGS Guilt and SSGS shame during the guilt condition did not differ significantly from those in the control condition. This should be considered as more than evidence that guilt and shame were both present in the guilt condition (lines 217-221).

---

## Round 0.2 · accepted · Accept

Excellent work that we are pleased you submitted to PeerJ.

·

Basic reporting

In this study, the authors attempted to address a perceived gap in the literature on emotion and decision-making by inducing guilt (the effects of which are documented) and joy (the effects of which were unclear) in subjects and examining their behavior in the Prisoner's Dilemma (PD). The sought to examine whether the effect of these emotions was significant after controlling for subjects' pre-existing tendency to cooperate (measured via a social value orientation task). They hypothesized that the effect of induced emotion would be significant. To do so, they first conducted a study validating their emotion induction procedure (guilt, joy or neutral emotion). Next, subjects were randomly assigned to one of each of these emotions for induction and made to play a repeated PD to assess the effect of the emotional condition on cooperation.

The authors found that while the emotion induction procedures were successful, their effect was not significant with regard to the subject's pre-existing tendency to cooperate (contrary to their hypothesis). However, there was a small but significant effect which emerged when the authors examined the initial trials following the induction. The authors conclude with an interesting discussion of possible reasons for this largely negative finding and the implications for their conclusions.


The structure, introduction and background supplied are all sufficient for understanding the rationale for their design and analyses.

Experimental design

Good. The first study's validation-focused design lends especial credibility to the findings discussed in the second study as well as their interpretation.

Validity of the findings

No Comments.